# The Role of Transmembrane Proteins in Plant Growth, Development, and Stress Responses

**DOI:** 10.3390/ijms232113627

**Published:** 2022-11-07

**Authors:** Yingli Zhou, Baoshan Wang, Fang Yuan

**Affiliations:** Shandong Provincial Key Laboratory of Plant Stress, College of Life Sciences, Shandong Normal University, Ji’nan 250014, China

**Keywords:** development, transmembrane proteins, stress response, structures

## Abstract

Transmembrane proteins participate in various physiological activities in plants, including signal transduction, substance transport, and energy conversion. Although more than 20% of gene products are predicted to be transmembrane proteins in the genome era, due to the complexity of transmembrane domains they are difficult to reliably identify in the predicted protein, and they may have different overall three-dimensional structures. Therefore, it is challenging to study their biological function. In this review, we describe the typical structures of transmembrane proteins and their roles in plant growth, development, and stress responses. We propose a model illustrating the roles of transmembrane proteins during plant growth and response to various stresses, which will provide important references for crop breeding.

## 1. Introduction

Approximately half of the components of the lipid cell membranes of animal cells are proteins. Proteins are indispensable for all life activities and have critical functions in cell membranes [1,2]. Transmembrane proteins have unique roles, such as controlling signal transmission, transport of nutrients and other soluble molecules, and energy conversion across organelles and cells [3]. Recent advances in protein trafficking have allowed researchers to identify and utilize many transmembrane proteins in animals, especially in drug research and cancer treatment [4,5]. Furthermore, the rapid development of plant genome sequencing technologies and molecular analyses have helped identify and predict transmembrane proteins in plants [6].

About 20–30% of genes are predicted to encode transmembrane proteins in current genome databases [7], but only 1% of entries in the Protein Data Bank are transmembrane proteins [8], suggesting that many transmembrane proteins have not been functionally described or analyzed.

While exploring the influence of transmembrane proteins on the formation of hydrophilic channels in cells, researchers discovered that membrane proteins tend to be more polar and hydrophilic than the hydrophobic tails of phospholipid molecules, and thus affect nearby phospholipid molecules [9,10]. For example, transmembrane proteins cross the cell membrane, which will affect the thermodynamic properties and dynamic structures of their neighboring phospholipids [11]. Phospholipid turnover often occurs in areas of the cell membrane that are disturbed by transmembrane proteins or other foreign substances [12,13]. Some transmembrane proteins form or facilitate the formation of hydrophilic channels in cell membranes and reduce transmembrane potential, thus improving the efficiency of substance transport [14].

The functional study of transmembrane proteins has accelerated their utilization in therapeutics. For example, members of the G protein family of transmembrane proteins are used as drug targets [15]. As transmembrane protein-mediated signal transduction, G protein-coupled receptor plays an important role in nervous system development, energy metabolism regulation, glucocorticoid receptor signals, and other physiological activities [16]. Researchers recently identified a transmembrane protein, CUB-containing domain protein 1 (CDCP1), as a promising therapeutic target and diagnostic biomarker for malignant tumors [17].

Progress in discovering transmembrane proteins has also accelerated in plants [18]. For example, the transmembrane protein 14 family comprises proteins containing the Tmemb-14 transmembrane domain. A member of this family, FATTY ACID EXPORT 1 (FAX1), is implicated in fatty acid transport in Arabidopsis (*Arabidopsis thaliana*), a dicotyledonous model plant [19]. In addition, TMEM165d localizes in the plasma membrane and regulates intracellular ion homeostasis by transporting magnesium (Mg^2+^) during pollen growth and development in Arabidopsis [20].

Transmembrane proteins have complex structures, particularly in their transmembrane domains, which vary widely between proteins, making their functions and relationships challenging to study [21,22]. Furthermore, it is not well understood how transmembrane proteins affect phospholipid molecule turnover or hydrophilic channel formation [23]. It is difficult to maintain the activity of purified transmembrane proteins in vitro; therefore, artificial vesicles have become the go-to method to study the structure and function of transmembrane proteins [24]. Here, we review the structures, and physical and chemical properties of transmembrane proteins, as well as their roles in plant growth and development, signal transmission, and stress responses, to provide a basis for further study of transmembrane proteins.

## 2. The Structure of Transmembrane Proteins

Transmembrane proteins can be divided into two main types based on the structure of their transmembrane domains: those with a transmembrane α-helix (TMH) structure, and those with a transmembrane β-barrel (TMB) structure [25]. With the exception of the β-barrel structure of transmembrane proteins in the outer membranes of bacteria and mitochondria, most other transmembrane proteins are α-helical [26]. Hydrophobic fragments with one or more α-helix structures form a transmembrane structure, and an α-helix structure contains about 15–25 amino acids [27]. Transmembrane proteins interact with the lipid bilayer and the water interface via hydrogen bonding, hydrophobicity, and bipolarity. The lipid bilayer mediates protein activity by influencing the structure of membrane proteins [28].

The amino acid sequence of a protein will determine its localization within the membrane and how it will interact with other helices [29]. Hydrophobic amino acid residues tend to form transmembrane structures, while polar and ionizable amino acid residues, such as proline, tend to occur at/in the water interface [30]. However, polar and ionizable amino acids also exist in many transmembrane structures and play an important role in protein structure and function. Polar amino acids can reduce the energy required by hydrogen bonding, while ionizable amino acids can reduce the energy consumption needed for embedding in the lipid bilayer by forming non-ionized states or salt bridges [31].

Most transmembrane proteins cross the lipid bilayer via one or multiple α-helix domains [32]. In these proteins, the hydrophobic region interacts with the hydrophobic tails of lipid molecules in the lipid bilayer, while the hydrophilic regions are located on both sides of the membrane [31]. Proteins that cross the lipid bilayer via a single α-helix are called single transmembrane proteins (Figure 1A). Many of these proteins function as signal receptors [33]. Some membrane proteins reside only on the cytoplasmic side of the membrane, with the α-helix structure associated with the internal lipid layer (Figure 1B). Other membrane proteins pass through the lipid bilayer via several α-helix domains, and are called multispanning transmembrane proteins (Figure 1C). An α-helix structure that completely crosses the membrane contains at least 20 amino acids. Therefore, the amino acid sequence of a protein, and in particular the presence of α-helices (which generally consist of 20–30 highly hydrophobic amino acids), is used to predict the presence of a transmembrane domain. Furthermore, X-ray crystallography has been used to identify the three-dimensional structure of many membrane proteins [34]. Other ways in which transmembrane proteins bind to membranes include: via a β-barrel transmembrane structure [35]; the formation of large protein complexes; covalent binding to lipid groups; and noncovalent interactions with membrane proteins.

Transmembrane helices often interact or combine with other α-helices to form channels that allow water-soluble molecules to pass through the membrane [36]. These pore-forming proteins usually consist of a series of α-helices passing through the bilayer membrane. Many single channel membrane proteins form homolog or heterodimer [37]. These dimers are bound together by non-covalent but strong and specific interactions between two transmembrane helices. The hydrophobic amino acids of these helices guide protein-protein interaction [38]. In proteins with multiple transmembrane structures, transmembrane regions consist of helices containing hydrophobic and hydrophilic amino acid side chains [30]. To form a channel, these α-helices are arranged side by side in a ring shape, with the hydrophobic side chains associated with the membrane side, and the hydrophilic side chains lining the inside of the hydrophilic pore [39].

Some β-folded sheets cross the membrane several times to form large, cylindrical ion channels called β-folded barrels (Figure 1D) [40]. The number of β-folded sheets varies, with as few as eight and as many as 22. The amino acid side chains facing inside the barrel are mainly hydrophilic, while those outside the barrel that contact the lipid bilayer are completely hydrophobic. Unlike the α-helix, the β-folded barrel can only form a wide channel because the tightness of the β-folded sheets bent into the barrel is limited [41], meaning it is is not as flexible as the α-helix.

Transmembrane proteins are not stable in vitro [42], making them difficult to study, and most transmembrane protein structures can only be predicted by computer algorithms [43].

## 3. Methods Used to Study Transmembrane Proteins

Although researchers can predict transmembrane proteins through genomic research, the specific structural information of the transmembrane region is still unclear, which requires the use of algorithms to predict the transmembrane region and transmembrane direction, so as to guide the study of transmembrane proteins [44]. The transmembrane protein prediction algorithm is an algorithm specifically for predicting the number of transmembrane fragments in a protein sequence. Based on a large amount of data analysis on the transmembrane region of transmembrane protein, especially on the topological structure of transmembrane protein [45], how to solve the problem of membrane protein structure prediction using bioinformatics was investigated. The detailed information of a protein sequence including several transmembrane fragments, the starting and ending positions of each transmembrane fragment, and some correlation coefficients were obtained by using the prediction algorithm [46].

Five methods were used to predict transmembrane protein structure: (1) the classical Kyte and Doolittle method [47]; (2) the “positive charge residence rule” [48]; (3) hidden Markov models [49]; (4) statistical analysis and model parameters [50]; and (5) wavelet analysis [51]. Each prediction method has unique requirements, and each relies on efficient and accurate algorithms.

### 3.1. Prediction by the Classical Kyte and Doolittle Method

The earliest and simplest prediction algorithm for transmembrane proteins was reported by Kyte et al. in 1982, who used the hydrophobic free energy of amino acids to convert the protein sequence into a hydrophobicity map [47] and then set a suitable threshold to predict the transmembrane region. However, this method may not be suitable for multiplanar channel proteins, with other methods potentially being more suitable for such predictions.

### 3.2. Prediction Using the “Positive Charge Residence Rule”

In 1986, Von Heijne [48] proposed the positive charge interior rule, which greatly improved the accuracy of transmembrane protein prediction. Rost et al. [52] integrated the sequence alignment of multispanning transmembrane proteins into a neural network with the positive charge interior rule to predict transmembrane regions. 

### 3.3. Prediction Based on Hidden Markov Models

Sonnhammer et al. [49] proposed a hidden Markov model-based method, which used a model of seven states, each corresponding to different regions of the transmembrane protein, namely the transmembrane core and the two sides of the transmembrane region, transmembrane ends, loops within the membrane, short and long loops outside the membrane, and regions away from the membrane.

### 3.4. Statistical Analysis and Model Parameter Prediction

Two algorithms, MEMSAT and TMAP, take into account the preference of amino acid distribution [53] to predict the structure of transmembrane proteins. Statistical analysis by MEMSAT [50] is used to obtain the frequencies of amino acids appearing in the inner membrane, outer membrane, transmembrane core region, and transmembrane terminal region, as well as their frequencies in the entire transmembrane protein sequence. The ratio of these two frequencies is used to predict the likelihood of amino acids appearing in any one of these regions, and the transmembrane structure of the target sequence is predicted by a dynamic algorithm. The prediction methods reported by Jones et al. [54] and Persson and Argos [55,56] are also based on residue occurrence frequencies, and they can also predict the transmembrane direction. Hydrophobicity is the main feature of the transmembrane helix, and long stretches of hydrophobic amino acids also exist in the hydrophobic core of globular proteins, which will produce a false-positive prediction result [57].

The TMAP method [58] is based on the statistical analysis of a transmembrane protein family, and predictions are based on protein sequence alignments. The prediction softwares TMHMM and HMMTOP [59,60], which are based on a hidden Markov model, predict protein structure through model parameters and algorithms, which are supported by thorough mathematical theory, meaning that both methods have high prediction accuracy. Therefore, TMHMM, HMMTOP, and MEMSAT predict protein structures according to single-protein sequence information, while TMAP and PHDhtm [61] predict protein structures according to multiple-protein sequence alignments, which account for sequence conservation among proteins and thus tend to overcome the noise inherently caused by sequence variation and improve prediction accuracy.

### 3.5. Prediction by Wavelet Analysis

The application of wavelet analysis [51] in protein structure studies began in 1998. Using bioinformatics, Hirakawa et al. [62] predicted hydrophobic amino acids in proteins using multi-resolution analysis of wavelet transform. Later, Arnold et al. [63] used continuous wavelet transform to reveal the similarities and differences between protein structures. Murray et al. [64] also employed continuous wavelet transform to describe and detect the active site of proteins. Qiu et al. [65] used single-scale continuous wavelet transform to predict the transmembrane domain(s) of membrane proteins. In addition, the continuous wavelet transform method can be used to predict the connecting polypeptides (i.e., partial regular and irregular secondary structures) between the the α-helix and the β-folded [66]. Wavelet analysis is known as a “mathematical magnifier” when conducting multi-scale, detailed analyses on the signal (function) through scaling and smoothing. Therefore, by using wavelet analysis, the local characteristics of the signal in the time and frequency domains can be obtained at the same time. For the study of non-stationary signals and mutation signals, wavelet analysis has natural advantages [67].

The distribution of amino acids and their effect on protein structure are also related to the length [68] and hydrophobicity gradient of transmembrane regions [69]. Therefore, to eliminate noise and improve prediction accuracy, various influencing factors should be taken into account. Comparing the precision of different algorithms for predicting transmembrane protein structure revealed the critical role that protein structure plays in protein function, and provided a basis for future studies on the biological functions of transmembrane proteins [70].

## 4. The Role of Transmembrane Proteins in Plant Growth and Development

Transmembrane proteins lie at the interface between cells and the external environment, and have many important functions [71]. For example, transmembrane proteins participate in the exchange and transmission of phytohormones, signaling molecules, and various substrates and metabolites between cells [72]. Many transmembrane proteins are ion channels [73] that transport or remove ions and toxic molecules out of cells. Transmembrane proteins have many other functions: some fix the membrane on the actin cytoskeleton on both sides [74]; some detect chemical signals in the surrounding environment and transmit them to the cell interior [75]; and some function as enzymes [76]. Each type of cell membrane contains different proteins, which perform specific functions.

**Leaf development**. Transmembrane proteins in plants are responsible for the uptake and transport of ions, molecules, and phytohormones, and are indispensable in plant growth and development [77]. 

Transmembrane proteins function in leaf development. For example, the maize (*Zea mays*) protein ZmMATE884 [78] shares high sequence identity with members from the Arabidopsis MATE (Multidrug And Toxic Compound Extrusion) transporter family [79]. The expression levels of *ZmMATE884* gene in the upper and lower leaves of maize were high, indicating that the gene might be involved in the regulation of the growth and development of leaves during the growth of maize [80]. The transmembrane protein aquaporin localizes to the tonoplast [81] and mediates water transport across the tonoplast to regulate cell turgidity [82]. *PgTIP1* (*Tonoplast intrinsic protein 1*) in ginseng (*Panax ginseng*) encodes an aquaporin and is abundantly expressed in leaves [83]. Heterologous expression of *PgTIP1* in soybean (*Glycine max*) significantly increases leaf growth [83]. Heterologous expression of *PgTIP1* in rapeseed (*Brassica napus*) and Arabidopsis [84] resulted in accelerated growth and a significant increase in leaf cell size.

**Root development**. Transmembrane proteins are also involved in root development. *ZmMATE884* is also highly expressed in maize roots, indicating a role in root growth and development [78]. 

Arabidopsis *ALF5* (*ABERRANT LATERAL ROOT FORMATION 5*) participates in lateral root development [85]. Additionally, *PgTIP1* is highly expressed in ginseng roots [86]; *PgTIP1* overexpression significantly promotes root growth [83]. Furthermore, heterologous expression of *PgTIP1* in rapeseed and Arabidopsis [87] accelerated growth and significantly increased root length.

**Reproductive growth**. Transmembrane transporters and transport mechanisms in Arabidopsis are popular research topics. About 13% of Arabidopsis genes encode transporters, yet their functions have not been fully elucidated [88].

GABA (gamma-aminobutyric acid) signaling pathways play an essential role in the regulation of reproductive growth and the responses to various stresses. GABA rapidly accumulated in plant tissues, after whicb the anion flux through plant aluminum-activated malate transporter (ALMT) was activated by anions and negatively regulated by GABA. In addition, this signal pathway stimulated by GABA and ALMT also regulated the growth of pollen tubes [89], which provides an important attempt for the participation of transmembrane proteins in the reproductive growth of plants. Arabidopsis FAX1 is a transmembrane protein located in the chloroplast inner envelope and transports fatty acids out of plastids [90]. Biomass and pollen fertility in the *fax1* mutant are significantly reduced relative to the wild type. In addition, lipid biosynthesis is strongly reduced, and the mutant plants have a short stature, pollen exine development defects, and decreased synthesis of leaves and flowers [86]. These results indicate that the loss of FAX1 affects Arabidopsis growth and development. *ZmMATE884* is highly expressed in maize embryos, supporting its involvement in embryo development [78]. Heterologous expression of *ZmMATE884* in Arabidopsis accelerates growth and hastens flowering, indicating its broad role in plant growth and development [80]. However, in the maize MAME family, only *Bige1* (*Big embryo 1*) was reported to be involved in maize organogenesis and organ size [91]. A detailed study of *PgTIP1* in ginseng revealed that *PgTIP1* overexpression promotes water and ion uptake, as well as photosynthesis. Furthermore, PgTIP1 regulates the expression of cell-cycle-related genes to accelerate cell growth [87]. 

**Phytohormone regulation**. Transmembrane proteins also function in plant hormone regulation [92]. As the earliest discovered hormone, auxin played a central role in regulations of plant growth and stress responses. Two important transmembrane proteins (PIN and AUX family) are responsible for the polar transportation of auxin [80,93]. As a typical transmembrane protein, PIN protein has ten transmembrane helices, which control the output of auxin from the cytosol to the extracellular space. Cytokinin and the signaling regulate many aspects of plant development, including cell proliferation and differentiation, bud and root growth, and responses to biotic and abiotic stresses. In recent years, it has been proved that cytokinin receptors are mainly located in the membrane of endoplasmic reticulum (ER). Cytokinin receptors are integral transmembrane proteins with histidine kinase activity, with their hormone sensing domains located on one side of the membrane, while histidine kinase and receptor domains are always located on the other side [94]. The cytokinin signaling system indicated that transmembrane protein was of great significance for hormone regulation in plants. ZmMATE884 partially localizes to the Golgi apparatus/endosome, and negatively regulates hypocotyl elongation, suggesting that ZmMATE884 may affect phytohormone biosynthesis, transport, and response [95]. The expression of the Arabidopsis ABC transporter gene *WBC11* (*WHITE-BROWN COMPLEX HOMOLOG PROTEIN 11*) is regulated by light, and also promoted by Abscisic Acid (ABA) [96]. Auxin is synthesized in tissues with high cell division activity. Even at low concentrations, auxin regulates gene expression via transcription factors and transmembrane proteins that regulate auxin efflux and transport [97]. In *Nicotiana tabacum*, researchers identified a member of the natural resistance-associated macrophage protein (NRAMP) family [98]. NRAMPs are transmembrane transporters that localize to the plasma membrane [99] in rice (*Oryza sativa*); the tonoplast and plasma membrane [100] in soybean; and the plasma membrane, tonoplast, and Golgi apparatus [101] in Arabidopsis. The promoter of the tobacco *NRAMP* gene contains *cis*-acting elements involved in plant hormone responses, indicating its role in plant hormone regulation [102].

The transmembrane proteins involved in plant growth and development discussed here are summarized in Figure 2.

## 5. Transmembrane Proteins Participate in Various Stress Responses

Located at the interface between cells and the external environment, transmembrane proteins sense and respond to a variety of environmental stresses, including abiotic (salt, drought, and temperature stresses [103]) and biotic stresses (pathogens and insect pests [104]) [105].

## 6. Response to Salt Stress

Researchers identified a protein with four transmembrane domains, MaTET, in the cytoplasmic membrane of banana (*Musa acuminata*). *MaTET* expression is upregulated in banana roots under salt stress, suggestive of its involvement in the response to salt stress. Heterologous expression of *MaTET* in Arabidopsis via *Agrobacterium tumefaciens*-mediated transformation revealed that MaTET localizes to the leaf epidermal cell membrane and in the root cap. Furthermore, the transgenic Arabidopsis lines exhibited stronger salt tolerance than the wild type [106], indicating that MaTET is involved in the salt stress response. 

Cotton (*Gossypium hirsutum*) GhSARP1 (Salt-Associated Ring finger Protein) is a C_3_H_2_C_3_-type E3 ubiquitin ligase located in the plasma membrane, with in vitro E3 ligase activity [107]. *GhSARP1* expression is downregulated by salt. Transgenic Arabidopsis lines expressing *GhSARP1* are sensitive to salt stress during germination and in the post-germination stage, indicating that GhSARP1 negatively regulates ubiquitination-dependent salt stress response [107]. 

The Arabidopsis ubiquitin ligase STRF1 (SALT TOLERANCE RING FINGER 1) is located at the plasma membrane and intracellular corpuscles, and salt stress induces the expression of its encoding gene. Loss of AtSTRF1 function accelerates root endocytosis, alters the expression of genes related to the membrane transport system, enhances tolerance to salt ions and osmotic stress, and reduces reactive oxygen species accumulation under salt stress [108]. 

*PgTIP1* is abundantly expressed in ginseng stems and confers salt-stress tolerance in ginseng, as was reported in transgenic Arabidopsis plants heterologously expressing *PgTIP1* [83]. Transgenic *PgTIP1*-expressing Arabidopsis plants grow better than wild-type plants under salt stress, as evidenced by accelerated root growth and significantly enlarged leaves. Concurrently, heterologous *PgTIP1* expression also upregulates the expression of salt stress-related genes in Arabidopsis [86]. This study revealed that PgTIP1 has important roles in multiple aspects of plant growth and development [109].

## 7. Response to Drought Stress

*MaTET* is upregulated in banana roots under drought conditions, indicating its role in drought-stress response. Arabidopsis *CHY ZINC-FINGER AND RING PROTEIN 1* (*CHYR1*) is induced by ABA (Abscisic Acid) and drought, and is mainly expressed in vascular bundles and stomata. AtCHYR1 enhances plant drought tolerance by promoting ABA-induced stomatal closure and reactive oxygen species production [110]. The AtCHYR1 homolog in desert poplar (*Populus euphratica*), PeCHYR1, localizes to the nucleus and the endoplasmic reticulum, and the expression of its encoding gene is also induced by ABA and drought [111]. Similar to Arabidopsis, H_2_O_2_ production is significantly enhanced and stomatal opening is decreased in transgenic hybrid poplar overexpressing *PeCHYR1*. Furthermore, compared to the wild-type control, *PeCHYR1*-overexpressing transgenic lines showed higher sensitivity to exogenous ABA [112]. These results indicate that CHYR1 is involved in stomatal closure through ABA-induced H_2_O_2_ production, and has a vital role in drought tolerance [111]. 

Tobacco NtTIP1 is a tonoplast aquaporin with six transmembrane regions. *NtTIP1* is significantly upregulated in drought-tolerant materials and significantly downregulated in drought-sensitive varieties [113]. Under drought stress, *NtTIP1* expression might enhance membrane channel activity and promote water movement into cells or vacuoles to maintain the osmotic balance and enhance plant tolerance to water stress [114]. 

## 8. Response to Cold and Heat Stress

A novel transmembrane cold regulation protein COR413-PM1 was illustrated in Arabidopsis [115]. The differential abundance proteins (DAPs) related to freezing were mainly involved in the metabolism of fatty acids, sugars, and purines. Four proteins were involved in the cold response: fatty acid biosynthesis 1 (FAB1) participated in fatty acid metabolism and may affect the plasma membrane structure; Fructose kinase 3 (FRK3) and sucrose phosphate synthase A1 (SPSA1) played roles in glucose metabolism and may affect osmoregulation under freezing stress; and GLN phosphoribosyl pyrophosphate amide transferase 2 (ASE2) affected freezing tolerance through the purine metabolic pathway.

Three transmembrane ricin channel plasma membrane intrinsic proteins, (PIP) RcPIP (Plasma membrane intrinsic proteins) 1;3, RcPIP2;1, and RcPIP2;2 [116] of castor bean (*Ricinus communis*) participate in adaptation to cold temperatures. Low-temperature stress significantly downregulates *RcPIP2;1* expression, similar to the rice genes *OsPIP1;1*, *OsPIP2;4*, and *OsPIP2;5* [89]. Downregulation of these *PIP* genes under cold stress may promote stomatal closure to delay cell dehydration, and thus help the plants tolerate low temperature stress. By contrast, cold stress upregulates *RcPIP1;3* and *RcPIP2;2* expression, suggesting an antagonistic relationship between RcPIP1;3/RcPIP2;2 and RcPIP2;1, which offers insight into the regulation of cold adaptation mediated by ricin regulatory channel proteins [117]. 

*Brassica napus* BnTR1 is a C_4_HC_3_-type ubiquitin ligase with two transmembrane domains at its N terminus [118]. *BnTR1* overexpression and heterologous expression enhances heat-stress tolerance in *B. napus* and rice, respectively [119]. BnTR1 may promote heat-stress tolerance by regulating the activity of calcium channels via inducing the expression of the heat shock transcription factor and heat shock protein genes [120]. 

## 9. Responsive to Biotic Stresses

Transmembrane proteins are also involved in the response to biological stresses. For example, plants recognize pathogens can activate defense responses through a large number of immune receptors. A few of these receptors were belonged to the receptor-like protein family such as SNC2 (for Suppressor of NPR1, Constitutive2). In Arabidopsis, the transmembrane protein BDA1 (for bian da, “becoming big” in Chinese) has been found to be a key signaling component downstream of SNC2. When SNC2 detects the unknown PAMP (pathogen-associated molecular pattern), it activates the transmembrane alkane protein BDA1 [121]. Once activated, BDA1 may recruit additional signaling components through its kinin repeat domain to activate SA (salicylic acid) synthesis and WRKY70-dependent defense gene expression, thereby enhancing plant defense responses.

In addition, the Arabidopsis homologous proteins SYNTAXIN OF PLANTS 121 (SYP121) and SYP122 are located at the plasma membrane, and the higher expression levels of their encoding genes causes cell wall thickening and mastoid formation to promote powdery mildew resistance [122]. 

Rice PENETRATION 1 (OsPENl), a homolog of Arabidopsis PEN1 and barley (*Hordeum vulgare*) HvROR2 (required for mlo-specified resistance2), contains a transmembrane domain. Transgenic rice lines overexpressing OsPEN1 exhibited enhanced resistance to *Magnaporthe grisea*, *Rhizoctonia solani*, and *Rhizoctonia solani*, while OsPEN1 RNA interference plants were more susceptible [123]. The OsPEN1 gene showed resistance to a variety of pathogens, which might be related to the non-host resistance of plants, and it was found that the protein Qa-SNARE enhanced the disease resistance of plants by participating in the disease resistance signaling pathway [123]. 

The transmembrane proteins involved in stress responses discussed here are summarized in Figure 3.

## 10. Future Perspectives

The functional characteristics of transmembrane proteins differ according to their cellular and tissue localization. Table 1 lists the classifications and roles of the current transmembrane proteins involved in plant growth, development, and stress responses. Protein discovery and functional characterization will facilitate future studies of the physiological and biochemical processes that occur in plant cells and membrane systems [124]. The following are important research topics that need to be addressed.

## 11. Application of Transmembrane Protein in Crop Breeding

With the discovery of more and more transmembrane proteins, the function of transmembrane proteins has attracted growing attention. In the growth processes of plants, transmembrane proteins not only help to promote the root, leaf, and reproductive development of plants, but also play an important role in plant resistance to biological and non-biological stress. In addition, studies have shown that transmembrane proteins also play a role in plant breeding. For example, the *GS3* gene in rice has a typical transmembrane domain and has been shown to contribute to grain length and weight in rice [125]. More investigation is neededof transmembrane proteins related to crop breeding based on the functional predictions and discovery of the functions of key transmembrane genes in order to improve crop yield.

## 12. Receptor Proteins for Various External Stimuli

Cell membranes contain many types of transmembrane proteins, which have important physiological functions, such as “carriers” to transport substances into and out of cells, and receptors for phytohormones or other chemicals [126]. 

Receptors are the key to any biochemical signal. Receptor-like kinases (RLKs) are transmembrane proteins that function as receptors of specific ligands, and are also protein kinases [127] in eukaryotes. They can establish signaling pathways to transfer the information crossing the cell membrane from outside to the nucleus and finally activate to guide the process of growth, development, stress response, and disease resistance. Multiple RLKs were identified in Arabidopsis to participate in the regulation of various physiological effects through different signaling pathways [128]. RLKs in plants participate in various physiological and biochemical reactions such as stress responses, morphogenesis, self-incompatibility, and growth and development. For example, Arabidopsis *ERECTA* encodes a leucine-rich repeat receptor-like protein kinase (LRR-RLK) that is associated with organ formation and vegetative growth [129]. In addition, the rice *Ralstonia solanacearum*-resistance gene, *Xa21*, encodes a protein with a typical LRR-RLK structure [130], indicating that plant cells can perceive and transmit external pathogen signals through RLKs located at the plasma membrane. Furthermore, BRASSINOSTEROID-INSENSITIVE1 (BRI1) is a major brassinosteroid receptor in plants [131]. BRI1 belongs to the LRR-RLK family, and is a typical single-transmembrane protein. The LRR domain of BRI1 is the site for brassinosteroid recognition. BRI1 regulates stem and root growth, vascular development, light development, and pollen tube elongation by recognizing and transmitting brassinosteroid signals [132].

The LRR-RLK family also participates in growth and development, stress resistance, and phytohormone signal transduction in plants [133,134]. In rice, the RLKs SIT1 (Signaling threshold regulating transmembrane adaptor l), SIK1 (Salt Inducible Kinase 1), and STRK1 (Salt Tolerance Receptor-like cytoplasmic Kinase 1) participate in the salt stress response [135]. To identify other RLK members that respond to salt stress, researchers investigated four *RLK* genes whose transcript levels were regulated by salt stress. Os4g0275100 was induced by NaCl, while the other three *RLK* genes were inhibited by NaCl. These four *RLK* genes had tissue-specific expression patterns: Os04g0275100 and Os07g0541900 were more expressed in roots, Os09g0353200 was mainly expressed in leaves, and Os01g0852100 was expressed in multiple tissues and organs [136]. However, whether these genes function as salt receptors remains to be investigated.

## 13. Interactions between Transmembrane Proteins and Intracellular Proteins

Transmembrane proteins participate in stress signal transduction and intracellular signal transduction in plants. Heterotrimeric GTP-binding proteins are key regulators of numerous signaling pathways in all eukaryotes. Heterotrimeric G protein (hereinafter referred to as G protein) refers to the core protein complex consisted of a Gα, a Gβ, and a Gγ subunit, and the activity was determined by the binding difference of guanine nucleotide and Gα. G protein is a typical transmembrane protein involved in intracellular signal transduction, and the existence of three subunits of G protein was closely related to the regulation of plant growth, development and maintenance of the normal phenotype [137]. Signal transduction and corresponding signaling pathways are achieved by the interaction of the three subunits and downstream effectors, and they also play an important role in seed yield of plants, organ size regulation, and biological and non-biological stress responses [137,138].

For example, in the G protein-mediated signaling pathway, Ca^2+^—as a secondary messenger—participates in signal transduction by binding to the conserved EF-hand structure of various Ca^2+^-sensing proteins [139] to modulate stress-responsive genes [140]. Several Ca^2+^-signaling pathway-related proteins are involved in the Ca^2+^ balance during the stress response, such as calmodulin and calreticulin. These Ca^2+^-binding proteins transmit Ca^2+^ signals to modulate the activities of specific downstream protein kinases to cope with the stress.

In plants, G proteins and small G proteins are involved in signaling pathways to respond to a variety of stresses (phytohormones, drought, ozone, and pathogens), serving as molecular switches [141]. G proteins sense and convert extracellular signals by interacting with receptors to activate enzymes [142]. G protein can interact with activated G protein-coupled receptors to directly sense and convert low temperature signals to cope with cold stress.

The phosphoinositide signaling pathway participates in plant growth, development, and stress responses. Cell membrane receptors receive extracellular signals, which then activate phosphoinositide-specific phospholipase C (PI-PLC) to hydrolyze phosphatidylinositol 4,5-bisphosphate to generate the messengers 1,4,5-inositol triphosphate (IP_3_) and diacylglycerol (DAG) [143]. PLC participates in various signaling pathways by regulating IP_3_ and DAG accumulation. IP_3_ stimulates extracellularly stored Ca^2+^ to enter the cytoplasm matrix, thus increasing the intracellular Ca^2+^ concentration, which further regulates the signal transduction pathway [144]. DAG is rapidly phosphorylated by diacylglycerol kinase to phosphatidic acid, which acts as a secondary messenger [143].

Protein phosphorylation is important for signal transduction in plant cells. After receiving the stress signal from membrane receptors, proteins are reversibly phosphorylated, which cells utilize to amplify and transmit the signal [145]. Reversible phosphorylation is accomplished by protein kinases and reversed by phosphatases, and occurs mainly at serine, threonine, and tyrosine residues.

The mechanisms of receptors in intracellular signal transduction are not yet fully elucidated, including how receptors transmit the signal and the specific pathway in the cell. Protein sequence and functional analyses, as well as genetic methods, are used to determine the function of transmembrane proteins [146], and the increased availability of plant genomes and genetic resources has accelerated these studies. Forward and reverse genetic analyses of mutants have been used to study the biological functions of transmembrane proteins [147]. New transmembrane proteins are being discovered and their functions are being revealed [148]. More in-depth explorations of the structures, regulatory mechanisms, and functions of plant transmembrane proteins will aid in understanding their diverse roles in plant growth and development.

## Figures and Tables

**Figure 1 ijms-23-13627-f001:**
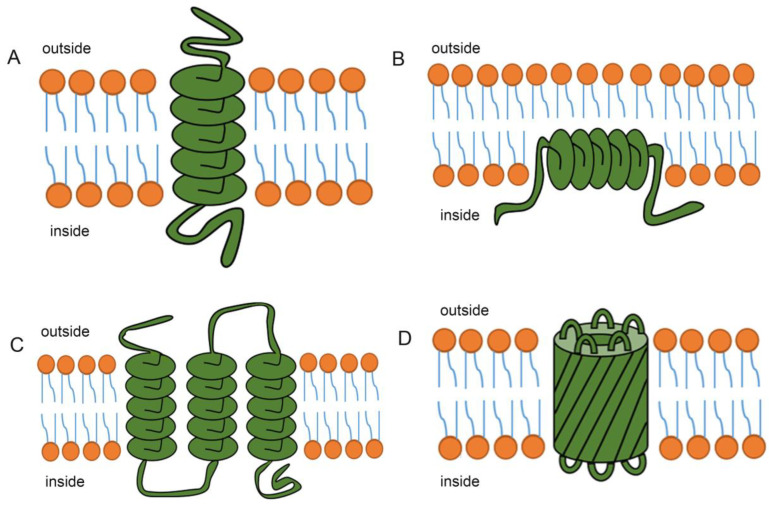
Structures of transmembrane proteins. (**A**) In a single-transmembrane protein, the polypeptide chain passes through the lipid bilayer via a single α-helix. (**B**) Some transmembrane proteins are almost completely located in the cytoplasm side of the lipid bilayer. (**C**) In a multi-transmembrane protein, the polypeptide chain passes through the lipid bilayer via two or more α-helixes. (**D**) Some transmembrane proteins transverse the plasma membrane via multiple β-folded sheets to form large ion channels.

**Figure 2 ijms-23-13627-f002:**
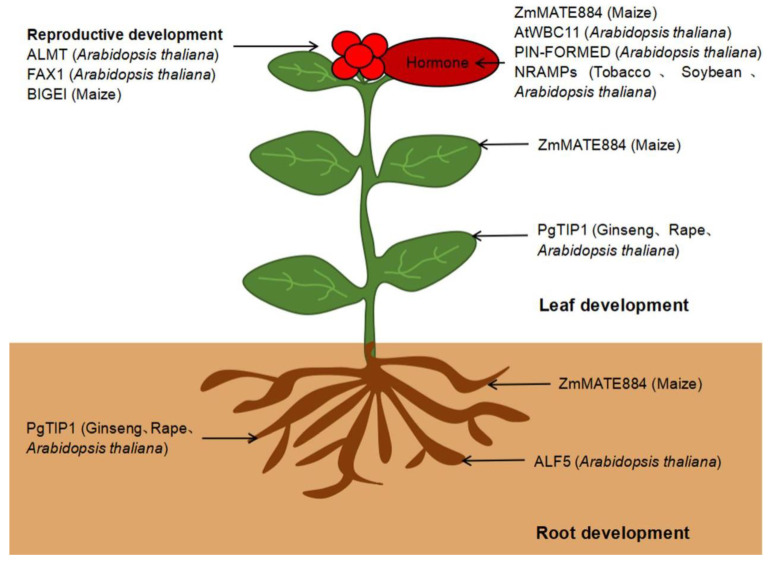
Transmembrane proteins involved in plant growth and development. ZmMATE884 in maize and PgTIP1 promote leaf growth in ginseng, *Brassica napus*, and Arabidopsis. ZmMATE884 in maize, ALF5 in Arabidopsis, and PgTIP1 in ginseng, *Brassica napus*, and Arabidopsis promote root growth. ALMT in Arabidopsis, FAX1 in Arabidopsis and BIGE1 in maize promote reproductive growth. ZmMATE884 in maize, AtWBC11 in Arabidopsis, PIN-FORMED in Arabidopsis, and NRAMPs in tobacco, soybean, and Arabidopsis participate in phytohormone regulation.

**Figure 3 ijms-23-13627-f003:**
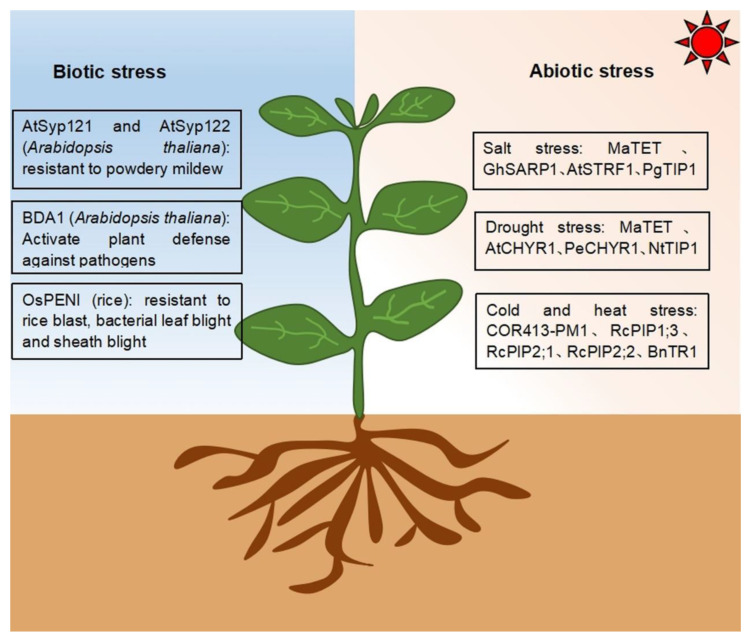
Transmembrane proteins involved in biotic and abiotic stresses. The homologs AtSYP121 and AtSYP122 in Arabidopsis are associated with powdery mildew resistance, BDA1 in Arabidopsis can activate plant defense responses to pathogens, and OsPEN1 confers resistance to rice blast, bacterial leaf blight, and sheath blight. MaTET, GhSARP1, AtSTRF1, and PgTIP1 in banana, cotton, Arabidopsis, and ginseng, respectively, promote salt stress tolerance. MaTET, AtCHYR1, PeCHYR1, and NtTIP1 in banana, Arabidopsis, desert poplar, and tobacco, respectively, promote drought-stress tolerance. COR413-PM1, RcPIP1;3, RcPIP2;1, and RcPIP2;2 in castor bean promote tolerance to cold stress. BnTR1 in *Brassica napus* promotes heat-stress tolerance.

**Table 1 ijms-23-13627-t001:** The transmembrane proteins involved in plant growth, development, and stress responses.

Classifications	Transmembrane Protein	Roles	Species
Leaf development	ZmMATE884	Participating in the growth and development regulation of the branches and leaves.	Maize [80]
	PgTIP1	Promoting the leaf size of panax ginseng and accelerating the growth of rape seed and *Arabidopsis thaliana.*	Ginseng [83], Arabidopsis, and rape [84]
Root development	ZmMATE884	Participating in the growth and development regulation of the root system of maize.	maize [78]
	ALF5	Participating in lateral root development.	Arabidopsis [85]
	PgTIP1	Promoting root growth.	Ginseng [83], Arabidopsis, and rape [87]
Reproductive growth	ALMT	Regulating the growth of pollen tubes.	Arabidopsis [89]
	FAX1	Enhacing the male fertility, lipid synthesis, and horn synthesis of leaves and flowers.	Arabidopsis [86]
	ZmMATE884	Participating in the growth and development regulation of the embryo after pollination.	Maize [80]
	BIGEl	Regulating organogenesis and organ size.	Maize [91]
	PgTIP1	Promoting photosynthesis and growth of plants.	Ginseng [87]
Phytohormone regulation	PIN and AUX family	Responsible for polar transport of auxin.	*Arabidopsis thaliana* [93,80]
	ZmMATE884	Regulating the development of plant hypocotyls by affecting synthesis, transportation, or response of plant hormones.	Maize [87]
	AtWBC11	Abscisic acid was involved in its expression to regulate hormones and light signals.	Arabidopsis [92]
	NRAMP	Participating in plant hormone regulation.	Arabidopsis, paddy, soybean, and tobacco [99]
Response to salt stress	MaTET	Participating in SOS signal transduction pathway under salt stress.	Arabidopsis and banana [103]
	GhSARP1	Negatively regulating ubiquitous-mediated salt stress response.	Arabidopsis and cotton [104]
	AtSTRF1	Promoting the tolerance to salt ion and osmotic stress.	Arabidopsis [105]
	PgTIP1	Improving the salt tolerance.	Ginseng and Arabidopsis [106]
Response to drought stress	MaTET	Participating in molecular responses.	Banana and Arabidopsis [103]
	AtCHYR1	Promoting ABA-induced stomatal closure and ROS production.	Arabidopsis [108]
	PeCHYR1	Inducing stomatal closure to improve drought resistance.	Poplar and Arabidopsis [109]
	NtTIP1	Enhancing channel activity to promote water absorption and maintain osmotic balance.	Tobacco [111]
Response to temperature stress	COR413-PM1	Regulating tolerance of freezing stress.	Arabidopsis [112]
	RcPIP1;3, RcPIP2;1, RcPIP2;2	A functional antagonistic relationship exists among the three genes to regulate cold adaptation.	Castor-oil plant [114]
	BnTR1	Regulating Ca^2+^ dynamics to induce heat shock protein expressions and promote the tolerance of heat stress.	Rape, rice [117]
Responsive to biotic stresses	BDA1	Activating SA synthesis and WRKY70-dependent defense gene expression to enhance plant defense responses.	Arabidopsis [118]
	AtSYP121, AtSYP122	Resistant to powdery mildew.	Arabidopsis [119]
	OsPENl	Disease resistance to a variety of pathogens.	Rice [120]

## Data Availability

Data sharing not applicable to this article as no datasets were generated or analysed during the current study.

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
