# Peer review of "The Role of Transmembrane Proteins in Plant Growth, Development, and Stress Responses"

_ijms, 2022, doi:10.3390/ijms232113627_

Round 1

Reviewer 1 Report

The review attempts to depict the role of transmembrane proteins in plant development and stress and explains one of the main problems in their functional characterization. However, I consider that it is too ambitious trying to exemplify everything known in several plant species and therefore lacks many references throughout all the sections showed in the review.

Some examples for distinct topics:

Yang, Y., Zhang, Y., Ding, P., Johnson, K., Li, X., & Zhang, Y. (2012). The ankyrin-repeat transmembrane protein BDA1 functions downstream of the receptor-like protein SNC2 to regulate plant immunity. Plant physiology, 159(4), 1857-1865.

Su, C., Chen, K., Ding, Q., Mou, Y., Yang, R., Zhao, M., ... & Xi, Y. (2018). Proteomic analysis of the function of a novel cold-regulated multispanning transmembrane protein COR413-PM1 in Arabidopsis. International Journal of molecular sciences, 19(9), 2572.

I strongly suggest delimiting the review topic or extensively searching for all important missing transmembrane proteins described as for example in: Jose, J., Ghantasala, S., & Roy Choudhury, S. (2020). Arabidopsis transmembrane receptor-like kinases (RLKs): a bridge between extracellular signal and intracellular regulatory machinery. International Journal of Molecular Sciences, 21(11), 4000.

Receptors known are missing, for example CK’s perception and other phytohormones:

Romanov, G. A., Lomin, S. N., & Schmülling, T. (2018). Cytokinin signaling: from the ER or from the PM? That is the question!. New Phytologist, 218(1), 41-53.

In figure two, the participation of transmembrane proteins in root and leaf development is summarized distinguishably. However, more detail is lacking in the topics concerning reproductive structures since they are not even represented in the schematic (with a flower for example).

Minor corrections

In phrase: Many transmembraen proteins are ion channels cite 72, correct to Transmembrane.

In phrase: The Arabidopsis MA

TE gene ALF5(ABERRANT LATERAL ROOT FORMATION 5)participates in lateral root development cite 85, is incorrectly separated by text formatting.

In phrase: The OsPENl gene showed resistance to a variety of pathogens, correct in the name of gene  the letter “l” by the number “1”

I consider that the format of the citations should be revised, since the names of the authors appear in different formats.

Author Response

Response to Reviewer 1 Comments

  1. The review attempts to depict the role of transmembrane proteins in plant development and stress and explains one of the main problems in their functional characterization. However, I consider that it is too ambitious trying to exemplify everything known in several plant species and therefore lacks many references throughout all the sections showed in the review.

Some examples for distinct topics:

Yang, Y., Zhang, Y., Ding, P., Johnson, K., Li, X., & Zhang, Y. (2012). The ankyrin-repeat transmembrane protein BDA1 functions downstream of the receptor-like protein SNC2 to regulate plant immunity. Plant physiology, 159(4), 1857-1865.

Su, C., Chen, K., Ding, Q., Mou, Y., Yang, R., Zhao, M., ... & Xi, Y. (2018). Proteomic analysis of the function of a novel cold-regulated multispanning transmembrane protein COR413-PM1 in Arabidopsis. International Journal of molecular sciences, 19(9), 2572.

Response: Thank you very much for the suggestions. Yes, we agree with you and we have already supplemented the topic-related articles on pages 12 to 14 and also given the corresponding statements on the response of transmembrane protein to cold stress and biological stress.

A novel transmembrane cold regulation protein COR413-PM1 were illustrated in Arabidopsis[116]. The differential abundance proteins (DAPs) related to freezing were mainly involved in the metabolism of fatty acids, sugars and purines. Four proteins were involved in the cold response including that fatty acid biosynthesis 1 (FAB1) participated in fatty acid metabolism and might affect the plasma membrane structure; Fructose kinase 3 (FRK3) and sucrose phosphate synthase A1 (SPSA1) played roles in glucose metabolism and may affect osmoregulation under freezing stress; and GLN phosphoribosyl pyrophosphate amide transferase 2 (ASE2) affects freezing tolerance through the purine metabolic pathway. The details are shown in lines 363-382 in pages 12-13.

Transmembrane proteins are also involved in the response to biological stresses. For example, plants recognize pathogens can activate defense responses through a large number of immune receptors. A few of these receptors were belonged to the receptor-like protein family such as SNC2 (for Suppressor of NPR1, Constitutive2). In Arabidopsis, the transmembrane protein BDA1 (for bian da, “becoming big” in Chinese) has been found to be a key signaling component downstream of SNC2. When SNC2 detects the unknown PAMP (pathogen-associated molecular pattern), it activates the transmembrane alkane protein BDA1[122]. Once activated, BDA1 may recruit additional signaling components through its kinin repeat domain to activate SA (salicylic acid) synthesis and WRKY70-dependent defense gene expression, thereby enhancing plant defense responses. The details are shown in lines 400-410 in page 13.

  1. I strongly suggest delimiting the review topic or extensively searching for all important missing transmembrane proteins described as for example in: Jose, J., Ghantasala, S., & Roy Choudhury, S. (2020). Arabidopsis transmembrane receptor-like kinases (RLKs): a bridge between extracellular signal and intracellular regulatory machinery. International Journal of Molecular Sciences, 21(11), 4000.

Receptors known are missing, for example CK’s perception and other phytohormones:

Romanov, G. A., Lomin, S. N., & Schmülling, T. (2018). Cytokinin signaling: from the ER or from the PM? That is the question!. New Phytologist, 218(1), 41-53.

Response: Sorry for the omission and thank you very much for the reminder. We have added the related references involved in plant hormone regulation and receptors in signal transduction.

Cytokinin and the signaling regulates many aspects of plant development, including cell proliferation and differentiation, bud and root growth, and responses to biotic and abiotic stresses. In recent years, it has been proved that cytokinin receptors are mainly located in the membrane of endoplasmic reticulum (ER). Cytokinin receptors are integral transmembrane proteins with histidine kinase activity, with their hormone sensing domains located on one side of the membrane, while histidine kinase and receptor domains are always located on the other side[94]. The cytokinin signaling system indicated that transmembrane protein was of great significance for hormone regulation in plants. The details are shown in lines 279-287 in page 10.

Receptors are the key to any biochemical signals. Receptor-like kinases (RLKs) are transmembrane proteins that function as receptors of specific ligands, and are also protein kinases[128] in eukaryotes. They can establish signaling pathways to transfer the information crossing the cell membrane from outside to the nucleus and finally activate to guide the process of growth, development, stress response and disease resistance. Multiple RLKs were identified in Arabidopsis to participate in the regulation of various physiological effects through different signaling pathways[129]. The details are shown in lines 463-469 in page 15.

  1. In figure two, the participation of transmembrane proteins in root and leaf development is summarized distinguishably. However, more detail is lacking in the topics concerning reproductive structures since they are not even represented in the schematic (with a flower for example).

Response: Good suggestions. We have modified figure 2 to indicate the reproductive structure theme with a flower and the related genes. And the legends are also revised.

  1. Minor corrections

In phrase: Many transmembraen proteins are ion channels cite 72, correct to Transmembrane.

Response: Done. We have corrected shown in line 229 in page 8.

In phrase: The Arabidopsis MA

Response: Done. We have deleted it.

TE gene ALF5(ABERRANT LATERAL ROOT FORMATION 5)participates in lateral root development cite 85, is incorrectly separated by text formatting.

Response: Done. We have corrected as “Arabidopsis ALF5ABERRANT LATERAL ROOT FORMATION 5participates in lateral root development[85]” shown in lines 254-255 in page 9.

In phrase: The OsPENl gene showed resistance to a variety of pathogens, correct in the name of gene  the letter “l” by the number “1”

Response: Done and sorry for the mistake. We have corrected as OsPENl shown in line 429 in page 14.

  1. I consider that the format of the citations should be revised, since the names of the authors appear in different formats.

Response: Done. We have made changes according to your suggestion.

Once again, thank you very much for your professional comments and suggestions. We earnestly appreciate your professional work and we hope that this revision will be deemed suitable for publication in International Journal of Molecular Sciences.

Reviewer 2 Report

In this review article, the authors introduce a number of plant transmembrane proteins that participate in plant growth, development and stress responses following summary of existing methods that predict transmembrane proteins from their amino acid sequences.  In my impression, the manuscript is concise and containing informative lists of such membrane proteins (summarized in Figs. 2 and 3), although it is better to include more descriptions on the usefulness of the prediction approaches (see below).

Major concern

It is not clear for me what is possible by using the prediction approaches referred in the manuscript.  They can identify essentially all transmembrane proteins from the genome but their functions are largely unknown?  Otherwise, accuracy of the prediction is still limited and they can only partially identify the candidate transmembrane proteins?  Different prediction methods should have different potential to adequately predict transmembrane proteins.  For example, the Kyte and Doolittle method might not be useful for multispanning channel proteins, whereas other approach may be better for such prediction.  Please revise the manuscript to include more explanations on this point.  Even if the TMAP method and wavelet analysis are used, they still include mistakes?  Such information will be useful for researchers studying plant TM proteins.

Minor concerns

1)    In the last part of Abstract, the authors refer to the potential use of information of the transmembrane proteins in crop breeding.  However, I did not find any comments on this point in the main text.  It is a very interesting topic, and I recommend revising the manuscript to include some fact and perspective on this topic, possibly in the Future perspective part of this review (e.g. is there any examples in which plant membrane proteins are used in the crop breeding?).

2)    Page 2, line 9 from the bottom, “…that mediate substance transport and signal transmission…” I only know animal G protein-coupled receptors that mediate signal transduction, and the receptors that mediate substance transport across membranes would be exceptional, although it might be popular in plants.  Please add explanation on this point or revise the description.

3)    Page 4, line 14 “multiple transmembrane proteins” ; page 6, line 1, “multiple membrane proteins” should be expressed as “multispanning transmembrane proteins” or “multi-spanning transmembrane proteins”

4)    Page 11, line 15, Please spell-out “ABA” in the text.

5)    Page 14, line 4 from the bottom, “…translate cold stress…” This sentence means that low temperature directly induce interaction between G proteins and activated G protein-coupled receptors?  Alternatively, other chemical/protein ligand induced by cold stress indirectly activates the receptor?  Please revise the manuscript to clarify this point.

6)    I noticed (probably) unnecessary characters often appeared in the references (for example, …BEN-TAL N J T I B S… in ref. 8).

Author Response

Response to Reviewer 2 Comments

In this review article, the authors introduce a number of plant transmembrane proteins that participate in plant growth, development and stress responses following summary of existing methods that predict transmembrane proteins from their amino acid sequences.  In my impression, the manuscript is concise and containing informative lists of such membrane proteins (summarized in Figs. 2 and 3), although it is better to include more descriptions on the usefulness of the prediction approaches (see below).

Response: Thank you very much for these suggestions. These advices are constructive and helpful to improve our manuscript. We carefully revised the manuscript accordingly. The main revisions are marked in blue in the revised version. We have also amended some details and have improved the manuscript according to all your comments. The detailed responses are listed one by one below.

Major concern

  1. It is not clear for me what is possible by using the prediction approaches referred in the manuscript. They can identify essentially all transmembrane proteins from the genome but their functions are largely unknown? Otherwise, accuracy of the prediction is still limited and they can only partially identify the candidate transmembrane proteins?  Different prediction methods should have different potential to adequately predict transmembrane proteins.  For example, the Kyte and Doolittle method might not be useful for multispanning channel proteins, whereas other approach may be better for such prediction.  Please revise the manuscript to include more explanations on this point.  Even if the TMAP method and wavelet analysis are used, they still include mistakes?  Such information will be useful for researchers studying plant TM proteins.

Response: Perfect suggestions. Thank you very much for these suggestions, and we have added explanations for the transmembrane protein prediction algorithm. Although researchers can predict transmembrane proteins through genomic research, the specific structural information of the transmembrane region is still unclear, which requires the use of algorithms to predict the transmembrane region and transmembrane direction in order to guide the study of transmembrane proteins.

The transmembrane protein prediction algorithm is an algorithm specifically for predicting the number of transmembrane fragments in a protein sequence. Based on a large amount of data analysis on transmembrane region of transmembrane protein, especially on the topological structure of transmembrane protein, how to solve the problem of membrane protein structure prediction using bioinformatics was investigated.

It can be obtained by the prediction algorithm, including the detailed information of a protein sequence, the starting and ending positions of each transmembrane fragment and some correlation coefficients. Most of these prediction methods are limited to the use of one or several significant features of transmembrane protein, and cannot comprehensively and comprehensively use a variety of potential useful information. So there will be still further improvement in its prediction ability to improve prediction algorithm or propose new algorithm.

The details can be found in line 137-149 in page 5 and 163-164 in page 6.

  1. Minor concerns

1)    In the last part of Abstract, the authors refer to the potential use of information of the transmembrane proteins in crop breeding.  However, I did not find any comments on this point in the main text.  It is a very interesting topic, and I recommend revising the manuscript to include some fact and perspective on this topic, possibly in the Future perspective part of this review (e.g. is there any examples in which plant membrane proteins are used in the crop breeding?).

Response: Good advice. Yes, we have added transmembrane proteins related to crop breeding in the revised version as a new section “Application of transmembrane protein in crop breeding” in perspectives. With the discovery of more and more transmembrane proteins, the function of transmembrane proteins has attracted growing attentions. In the growth processes of plants, transmembrane protein not only helps to promote the root, leaf and reproductive development of plants, but also plays an important role in plant resistance to biological and non-biological stress. In addition, studies have shown that transmembrane proteins also play a role in plant breeding. For example, the GS3 gene in rice has a typical transmembrane domain and has been shown to contribute to grain length and weight in rice[126]. More practices are needed to attempt of transmembrane proteins related to crop breeding based on the functional predictions and discovery of the functions of key transmembrane genes in order to improve the crop yield. The details are shown in lines 447-458 in pages 14-15.

2)    Page 2, line 9 from the bottom, “…that mediate substance transport and signal transmission…” I only know animal G protein-coupled receptors that mediate signal transduction, and the receptors that mediate substance transport across membranes would be exceptional, although it might be popular in plants.  Please add explanation on this point or revise the description.

Response: Done. Sorry for the inappropriate elaboration. We have modified the content of G protein-coupled receptor as transmembrane protein-mediated signal transmission. The details are showed in lines 51-54 in page 2.

3)    Page 4, line 14 “multiple transmembrane proteins” ; page 6, line 1, “multiple membrane proteins” should be expressed as “multispanning transmembrane proteins” or “multi-spanning transmembrane proteins”

Response: Done. We have replaced with “multispanning transmembrane proteins”. The details are showed in line 103 on page 4 and line 168 on page 6.

4)    Page 11, line 15, Please spell-out “ABA” in the text.

Response: Done. Abscisic Acid was supplemented in line 345 in page 12.

5)    Page 14, line 4 from the bottom, “…translate cold stress…” This sentence means that low temperature directly induce interaction between G proteins and activated G protein-coupled receptors?  Alternatively, other chemical/protein ligand induced by cold stress indirectly activates the receptor?  Please revise the manuscript to clarify this point.

Response: Sorry for the inappropriate statements. We have not made this clear and have already made changes. G protein can interact with activated G protein-coupled receptors to directly sense and convert low temperature signals to cope with cold stress. The details are showed in the lines 528-530 in page 16.

6)    I noticed (probably) unnecessary characters often appeared in the references (for example, …BEN-TAL N J T I B S… in ref. 8).

Response: Done. We have revised the citations according to your suggestions.

Once again, thank you very much for your professional comments and suggestions. We earnestly appreciate your professional work and we hope that this revision will be deemed suitable for publication in International Journal of Molecular Sciences.

Round 2

Reviewer 1 Report

The authors have greatly improved with an exhaustive search of known information about transmembrane receptors. And they have respectfully taken into account all recommendations made to them.

Specifically, I have a big question of ignorance. Based on the title of the paper: "The role of transmembrane proteins in plant growth, development, and stress responses", I would think that the authors would provide a state of the art of at least most of the known transmembrane proteins and the processes in which they are involved.

However, I consider that even though many examples have been shown in the review, some important references have been overlooked and specifically, I can refer for example to: Tena, G. PIN finally up. Nat. Plants 8, 725 (2022) or Ung, K.L., Winkler, M., Schulz, L. et al. Structures and mechanism of the plant PIN-FORMED auxin transporter. Nature 609, 605-610 (2022). Which I consider to be very pertinent for the purposes of their paper. 

One example in another section would be the importance of the aluminium-activated malate transporter (ALMT) in the GABA signaling pathway: Ramesh, S., Tyerman, S., Xu, B. et al.  GABA signaling modulates plant growth by directly regulating the activity of plant-specific anion transporters. Nat Commun 6, 7879 (2015), Given that GABA has already some described effects on plant development.

In other topic for example, on line 459, the authors provide a section on G proteins, but what about the latest of G proteins, for example: Patel, J.S., Selvaraj, V., Gunupuru, L.R. et al. Plant G-protein signaling cascade and host defense. 3 Biotech 10, 219 (2020). or Heterotrimeric G-Protein Signaling in Plants: Conserved and Novel Mechanisms, Sona Pandey, Annual Review of Plant Biology, 2019 70:1213-238

The authors, on the other hand, describe excellently and in a very reader-friendly way the known ways to predict the structure and possible functionality of the proteins from in silico models, but they do not provide at the moment all the references about transmembrane proteins. 

Would you consider it prudent to state perfectly in the body of the text that you only mention examples for each development or stress section that you have elaborated?

If not, would you consider it prudent to limit it in some other way, for example in the title of the text?

I respectfully leave this comments open in case you might consider that it could help you.

Minor comments:

I detected Typos about blanks in the text.

Author Response

Reponses:

Thank you very much again for these professional suggestions. These advices are constructive and helpful to greater improve our manuscript. We have carefully revised the manuscript accordingly. The main revisions are marked in red in the revised version (round 2). We have also amended Table 1 and some details, and have improved the manuscript according to all your comments.

First, we strongly agree with you that the statement of the known transmembrane proteins and the involved processes is necessary to act in cooperation with the title. So Table 1 (Pages 19-20) was supplemented in the revised version to give comprehensive statements of the most current reported transmembrane proteins including the roles and classifications.

Second, we appreciate your advice of the related literatures. We have carefully read these papers and cited the latest progresses involved in PIN-FORMED auxin transporter and the aluminium-activated malate transporter (ALMT) of the GABA signaling pathway. On the one hand, as the earliest discovered hormone, auxin played a central role in regulations of plant growth and stress responses. Two important transmembrane proteins (PIN and AUX family) are responsible for the polar transportation of auxin [98, 99]. As a typical transmembrane protein, PIN protein has ten transmembrane helices, which controls the output of auxin from the cytosol to the extracellular space. The details can be found in lines 280-285. On the other hand, GABA (gamma-aminobutyric acid) signaling pathway played an essential role in the regulation of reproductive growth and the responses to various stresses. GABA rapidly accumulated in plant tissues, then the anion flux through plant aluminum-activated malate transporter (ALMT) was activated by anions and negatively regulated by GABA. In addition, this signal pathway stimulated by GABA and ALMT also regulated the growth of pollen tubes[93], which provides an important attempt for the participation of transmembrane proteins in the reproductive growth of plants. The details can be found in lines 258-265.

Third, thanks for the good literatures about G-Protein. We have supplemented the corresponding contents. Heterotrimeric GTP-binding proteins are key regulators of numerous signaling pathways in all eukaryotes. Heterotrimeric G protein (hereinafter referred to as G protein) refers to the core protein complex consisted of a Gα, a Gβ and a Gγ subunit, and the activity was determined by the binding difference of guanine nucleotide and Gα. G protein is a typical transmembrane protein involved in intracellular signal transduction, and the existence of three subunits of G protein was closely related to the regulation of plant growth, development and maintenance  of the normal phenotype[144]. Signal transduction and corresponding signaling pathways are achieved by the interaction of the three subunits and downstream effectors, and they also play an important role in seed yield of plants, organ size regulation, biological and non-biological stress response[144, 145]. The details are shown in lines 475-485.

Finally, according to your advice, we have given the related references in the part of predicted methods of transmembrane proteins. In addition, in order to make more precise statements of transmembrane proteins involved in plant growth and stress responses, Table 1 was drawn to make more comprehensively display in the descriptions of the functions. Furthermore, we checked the spaces and mistakes in all parts of the manuscript.

Once again, thank you very much for your professional comments and suggestions. We earnestly appreciate your professional work and we hope that this revision will be deemed suitable for publication in International Journal of Molecular Sciences.

Reviewer 2 Report

The authors appropriately respond to my comments.  I have no further questions.

I noticed that two periods were included after a sentence in line 143 ( was investigated..).

Author Response

Response:Done. We have made changes here, please see line 146 for details.

Once again, thank you very much for your professional comments and suggestions. We earnestly appreciate your professional work and we hope that this revision will be deemed suitable for publication in International Journal of Molecular Sciences.